# Effect of Structural Differences on the Mechanical Properties of 3D Integrated Woven Spacer Sandwich Composites

**DOI:** 10.3390/ma14154284

**Published:** 2021-07-31

**Authors:** Lvtao Zhu, Mahfuz Bin Rahman, Zhenxing Wang

**Affiliations:** 1College of Textile Science and Engineering (International Institute of Silk), Zhejiang Sci-Tech University, Hangzhou 310018, China; mahfuz.zstu@gmail.com (M.B.R.); zhenxing@zstu.email.cn (Z.W.); 2Shaoxing Baojing Composite Materials CO., LTD, Shaoxing 312000, China

**Keywords:** industrial textiles, 3D integrated woven spacer sandwich composites, X-ray computed tomography, designing new products

## Abstract

Three-dimensional integrated woven spacer sandwich composites have been widely used as industrial textiles for many applications due to their superior physical and mechanical properties. In this research, 3D integrated woven spacer sandwich composites of five different specifications were produced, and the mechanical properties and performance were investigated under different load conditions. XR-CT (X-ray computed tomography) images were employed to visualize the microstructural details and analyze the fracture morphologies of fractured specimens under different load conditions. In addition, the effects of warp and weft direction, face sheet thickness, and core pile height on the mechanical properties and performance of the composite materials were analyzed. This investigation can provide significant guidance to help determine the structure of composite materials and design new products according to the required mechanical properties.

## 1. Introduction

The crucial advantages and vast potential applications of sandwich structure composites make them more desirable than conventionally produced composites. The cost-effectiveness, lightweight, durability, good design ability, high manufacturing efficiency, and many more qualities make these composites suitable for different sectors and different applications [1,2,3]. Research and development on composite materials and processes gained significance and popularity after the 1940s and are still popular for advanced composites [4]. The idea of sandwich structure composites first emerged in 1985 in both Katholieke Universiteit Leuven (Belgium) and the University of Stuttgart and MBB (Germany) [5,6,7]. Three-dimensional structure composites made of spacer fabric have excellent tensile, flexural, impact, and crash-resistance properties, which allow lightweight applications to replace composites of conventional structure [1,8].

Previous studies have shown that the mechanical performance of mono spacer fabric composite (without additional reinforcement) materials is related to the pile height, distribution density, and geometrical structures [5]. Zhang et al. studied the mechanical properties of 3D integrated woven spacer composites based on face sheet structure and found that face sheets with a complex structure and additional yarn arrangement have better warp direction bending and impact resistance properties relative to surfaces with a plain structure [9]. Shaokai Wang et al. compared 3D spacer fabric composites laminated with an additional glass woven face sheet and an innovative integral multiface sheet. They found that woven glass lamination strengthened the face sheet, while integral multiface sheet improved the properties related to the core piles of composites [10]. Although there are many advantages of the sandwich structure, the bonding between the face sheet and core materials is one of the major weak spots. Integrated woven core piles provide better skin core deboning resistance than sandwich structures of other core materials, such as honeycomb, foams, balsa wood, and others [11]. Neje and Behera reported that the core pile structure as well as different cell heights, widths, and opening angles significantly affected the failure loads and energy absorbing capacity of 3D integrally woven spacer sandwich composites [12]. Wang et al. compared the flat compression, shear, and flexural properties of 3D fabric sandwich composites and 3D fiber sandwich composites and found that 3D fabric composites had better flat compression and shear properties than 3D fiber composites when both had similar flexural properties [13]. The mechanical properties of 3D woven sandwich composites depend on the specific strength and the moduli of fibers and the matrix material. However, the mechanical properties and performance of any composite material are also greatly influenced by the fabric architecture, the geometrical arrangement of the fibers of the composite material, and the bonding of fibers with the resin matrix [14]. Although the reinforcement material plays a significant role in the load-bearing capacity of the composites while the resin matrix contributes a minor role, the resin matrix also has a significant effect on the mechanical properties of composites [15].

This study examined the mechanical properties and performance of 3D integrated woven spacer sandwich composites under various loading conditions and structural specifications. XR-CT (X-ray computed tomography) was used for the first time to visualize the microstructural details and fracture morphologies of 3D integrated woven spacer sandwich composites. No previous research has explored this method to analyze 3D integrated woven spacer sandwich composites. According to the literature, the performance and mechanical properties of composite materials vary according to their structural parameters. The advantages and limitations of composite materials also depend on them. Therefore, it is still necessary to study the composite structure to achieve the important features required for end-use materials.

## 2. Materials and Methods

### 2.1. Materials

For this study, three-dimensional integrated woven spacer fabric composite materials were provided by Sinoma Science & Technology Co., Ltd. (Nanjing, China. High-performance alkali-free E glass fiber produced by Jushi Group Co., Ltd. (Jiaxing, China) was used for the reinforcement fiber material. The chemical compositions included SiO, Al_2_O_2_, B_2_O_2_, TiO_2_, MgO, CaO, Fe_2_O_3_, Na_2_O + K_2_O, and F. The surface of the glass fiber was coated with a silane-based infiltrant suitable for weaving, winding, pultrusion, and other processes. The linear density of the glass fiber was 140 tex, and the fiber diameter was 13 mm. The reinforcement fabric is shown in Figure 1. The fabric structure was divided into upper and lower layers and Z-direction fibers. The Z-direction fiber warp surface presented an “8” shape, and the weft surface presented a “1” shape. The resin was Huibo New Material 5078 resin (epoxy resin), provided by Huibo New Material Technology (Shanghai, China) Co., Ltd. The specifications of the resin matrix were mixed viscosity at 25 °C, 200–300 cps, Tg 75–85 °C; bending strength of 100–120 MPa; and curing conditions of 50 °C × 4 h + 70 °C × 6 h. The 3D integrated woven spacer fabric specifications are given in Table 1.

#### Composite Material Molding Process

We used the hand lay-up process to prepare 3D integrated woven spacer sandwich composites. A schematic diagram of the hand lay-up process is shown in Figure 2.

First, a 100 cm × 100 cm glass plate was placed flat on the desktop and covered with an 80 cm × 80 cm film. Then, one-third of the prepared resin matrix was poured on the film and spread uniformly with a brush. The weight ratio of the resin to glass fiber was 1.1:1. Next, the cut reinforcement was spread on the film, and the remaining resin was poured on the matrix and spread evenly with a brush. The hollow layer fibers instantly absorbed the resin, and the Z-direction fibers had a “capillary action” effect. Under the automatic infiltration of resin, the fabric was formed automatically to the designed height. Finally, a film of the same size was spread on the fabric and solidified at room temperature to obtain a three-dimensional integrated woven spacer sandwich composite material board. The specifications of the prepared composite materials are shown in Table 2. It was noticed that after the molding process, the core pile height of the composite samples became lower than the core pile height of the fabric. Due to the resin matrix, the weight of the resin and the face sheet thickness resulted in the core pile height of the composite samples being lower than that of the actual fabric. The average measurements of five specimens of each group of samples are given in Table 2.

### 2.2. Mechanical Test

In this study, five different experimental analyses were conducted to analyze the mechanical properties, namely the three-point bending test, flatwise compression test, edgewise compression test, and the tensile test. XR-CT was used to analyze the fracture morphologies and visualize the microstructural details of 3D integrated woven spacer sandwich composites.

The size of the bending test sample was 150 mm × 50 mm. There were five warp and weft samples of each specification, making up a total of 50 samples. The schematic diagrams of the warp and weft direction samples are given in Figure 3a,b, respectively.

The bending experiment was conducted according to the standard GB/T 1449-2005 (test method for bending performance of fiber-reinforced plastics) [16]. The experiment was carried out under the conditions of one atmosphere pressure and a constant temperature of 25 °C. The loading speed was 2 mm/min, and the span was set to 110 mm. After placing the composite material sample board horizontally, the loading head was kept still. The bracket holding the composite material sample board rose at a constant speed and gradually destroyed the composite material sample board. The time displacement, loading load, bending strength, elastic modulus, and bending strain data were generated automatically in the Bluehill 3 system. Figure 4a shows the equipment set up for the bending experiment, and Figure 4b shows the equipment set up for the flat compression experiment.

At present, there is no unified test standard for testing the flat compression performance of glass fiber 3D spacer fabric composites in the industry. In this study, GB/T 1453-2005 “Sandwich structure or core flat compression performance test method” was selected based on the material characteristics and experimental equipment [17]. According to the standard, the test specimens were cut to a size of 60 × 60 mm. Five pieces were prepared for each group of standard specimen, as shown in Figure 5. The experiment was carried out on the INSTRON-8801 universal tester.

The effect of different core layer heights of 3D integrated woven spacer sandwich composites on their flat compression performance was explored. Two sets of samples were set up for comparison, as shown in Table 3.

The tensile test was carried out according to the standard ASTM D3039 [18]. Samples A, B, C, and D were selected from five different groups of samples. Two specimens were selected from each of these four groups of samples to test the weft tensile properties, and three specimens were selected from sample group A to test the warp tensile performance. Figure 6 shows the schematic diagram of the tensile test specimen.

The edgewise compression experiment was carried out according to the reference standard ASTM D3410 [19]. For the edgewise compression performance test, five different specifications were selected from groups A, B, C, D, and E. Two specimens from each group of samples were selected for the weft compression test. However, the group E sample was damaged due to the clamping failure during clamping and could not be tested. Therefore, only the compression curves of sample groups A, B, C, and D were obtained. Three specimens were selected from sample group C for the warp compression performance test. This experiment was carried out under standard atmospheric conditions of constant temperature and humidity (temperature 23 °C, relative humidity 50%). Before the experiment, the cut-up fabric samples were placed in a laboratory with constant temperature and humidity for 24 h. The experimental instrument was S4690A IITRI compression fixture. The compression tester was debugged, preparations were made for the experiment, and all the required parameters were set up for the test. Figure 7 shows a schematic diagram of the edgewise compression test specimen.

XR-CT experiment was employed to observe the fracture morphologies and visualize the microstructural details of tensile, edgewise compression, and flexural fractured specimens. For XR-CT sample preparation, samples of three different heights (3.40, 8.69, and 12.62 mm) were cut by a blade from the tensile, edgewise compression, and flexural fractured specimens. This experiment was performed with a high-resolution XR-CT machine named ZEISS Xradia 610 Versa, as shown in Figure 8.

## 3. Results and Discussion

### 3.1. Bending Performance

The original data were processed and stress–displacement curves drawn for each group in the latitude and longitude directions. The warp stress–displacement curve is shown in Figure 9, and the weft stress–displacement curve is shown in Figure 10. The flexural stress $\sigma_{f}$ can also be calculated by the following formula [20]:(1)$$\sigma_{f}=\frac{3FL}{2bh^{2}}$$
where, $\sigma_{f}$ (flexural stress, MPa) for group A warp sample 1 is 90.28 MPa, load *F* = 316.2365 N, span length *L* = 110 mm, the thickness of the sample *h* = 3.40 mm, and width of the specimen *b* = 50 mm.

As can be seen from Figure 9 and Figure 10, the bending strength of the three-dimensional integrated woven spacer sandwich composites decreased with the increase in core height and face sheet thickness in both the warp and weft directions. This was mainly due to the structure of the three-dimensional integrated woven spacer fabric, which was affected by the unevenness of the resin matrix. The core material of the three-dimensional spacer woven fabric presented an “8” shaped structure. The bending strength was higher at lower core pile height due to the “8” shaped structure being tightly attached and not apparent because of the action of the resin matrix. When the height of the core pile increased, the “8” shaped structure became isolated and visible, causing the structure of the warp yarn to bend. The resin matrix was not clogged in the spacer layer like in the low pile height structure, thereby causing the composite material to lose stability. As a result, the bending strength was relatively low. As can be seen in Table 4, the weft bending strength of the three-dimensional integrated woven spacer sandwich composite material was greater than that for the warp direction.

The average bending elastic modulus was also higher in the weft direction, as shown in Figure 11. The weft bending strength and elastic modulus of the three-dimensional spacer woven composite material was greater than the warp direction because of the difference in the warp and weft structures of the three-dimensional spacer woven fabric [21]. Weft yarns are more densely arranged than warp yarns when the length of the samples is the same, and weft yarns are in a slightly inclined vertical arrangement. In contrast, the warp yarns generally show a bending yield state, so the bending strength and antideformation ability of weft are more significant than those of warp. It is worth noting that the average elastic modulus decline in both the warp and weft directions became less when the face sheet thickness increased with the increase in core pile height. Therefore, it is clear that the face sheet thickness greatly influences the stiffness of three-dimensional integrated woven spacer sandwich composites.

As can be seen in Figure 12, the average failure deflection of groups A, B, and C gradually decreased in both the warp and weft directions with the increase in core pile height, while the face sheet thickness was the same. This shows that under the action of uniform load, materials with a large core height fracture faster, while materials with a low core height have better flexibility due to the structure and the effect of the resin matrix. In groups D and E, when the face sheet thickness increased, the samples became more capable of bearing loading head force with more minor fractures. Hence, the bending deflection did not decrease gradually with an increase in core pile height due to the increase in the face sheet thickness. Moreover, the warp direction samples had higher failure deflection than the weft direction samples because the “8” shaped structure in the warp direction had better flexibility than the “1” shaped structure in the weft direction.

For low core height samples, the upper and lower layers easily moved together and bore the effect of force, so the average failure deflection was high. For high core height samples, the upper and lower layers could not move together smoothly due to increased gap between the face sheets. Moreover, due to the flow of shear force in the core phase, the upper face sheet became the weak point. The force on the upper surface layer quickly reached the maximum, and fracture occurred due to contact with the loading head. The top face sheet generally bears the compressive load and the bottom face sheet bears the tensile load, while pile failure occurs due to the shear load [16,22,23].

Figure 13 shows the reverse view of damaged specimens in groups C and D. In the group C sample, there was apparent damage in both the top and bottom face sheets. The bottom face sheet fracture was more evident than the top face sheet fracture. However, in sample group D, the fracture was only seen in the bottom face sheet, with no apparent damage on the top face sheet. Due to the thickness of the top face sheet being higher than that of the bottom face sheet, the bottom face sheet was damaged first and the fracture was obvious. This observation clarifies that the thickness of the face sheet has a significant influence on the bending performance of the material. Different studies have shown that the bending load resistance of 3D integrated woven spacer composites can be improved using thickened face sheet. Complex-shaped face sheets have better performance than plain-shaped face sheets and experience minor damage on the bottom face sheet [9,24]. To observe the core pile damage phenomenon and visualize microstructural details of the bending, XR-CT test was conducted on the specimens. The core pile fracture and cross-sectional view of the group E sample can be seen in Figure 14. The XR-CT image showed the damage behavior of reinforced 3D integrated woven spacer fabric without the matrix material. The core pile yarn fracture and the yarn bending phenomenon were seen in the cross-sectional view of both the warp and weft directions. The internal torn fibers were attached to the resin matrix and looked undamaged in the specimen due to the resin matrix, but they were internally damaged and isolated, which was apparent in the XR-CT image.

### 3.2. Flat Compression Performance

Table 5 and Table 6 are the flat compression test data of sample groups A and D of three-dimensional integrated woven spacer sandwich composite material. The average value, standard deviation, and coefficient of variation are also given.

As the load increased, the sample compressed flatter and flatter. The compressive capacity of the material was continuously improved, so the strength limit of the flat compression sample could not be obtained. Therefore, the compressive load of the sample at yield was divided by the cross-sectional area of the sample, that is, the compressive strength limit σ (compressive strength) was used to express the compressive performance of the material. Data for groups A and D were compared, and it was found that when the height of the composite material decreased, the compressive strength of the material increased and the increase was apparent. The compression load-bearing capacity of 3D woven spacer fabric composites primarily depends on the pile height and resin matrix around the core piles [17]. Compressive stress at yield (MPa) was calculated based on the below formulas:(2)$$\sigma =\frac{F}{S}$$
(3)$$S=a^{2}$$
where σ refers to flat compression strength (MPa), *F* is the flat compressive yield load (*N*), *S* is the sample cross-sectional area (mm^2^), and *a* refers to the side length of the sample (mm). For group E samples, the load exceeded 100 kN. Because the maximum flat load of the test equipment was 100 kN, this experiment did not measure the specific compressive load. However, according to the formula of flat compression strength, the compressive strength was calculated as more than 25 MPa, which was more than four times higher than sample group D and 30 times higher than sample group A.

The load–time diagram (Figure 15) and the load–displacement diagram (Figure 16) of the experimental material were very similar. This was because the deformation of the sample was proportional to time, that is, the displacement in the experiment was proportional to time. The data of the two control groups in the flat compression test showed that the flat compression strength decreased with the increase in core layer height for the three-dimensional integrated woven spacer sandwich composites. Hence, it is clear that the flat compression properties of 3D integrated woven spacer sandwich composites decrease with the increase in core height. Comparing the flat compression data of 3D woven sandwich composites and 3D knitted sandwich composites, previous studies have shown that 3D knitted sandwich composites have better flatwise load-bearing capacity. In contrast, they have unacceptable load-bearing capacity in the edgewise direction [25,26]. Comparing the group B sample with top face sheet thickness of 1.10 mm and bottom face sheet thickness of 0.80 mm to the group C sample with top face sheet thickness of 0.75 mm and bottom face sheet thickness of 0.48 mm, the decrease in yield load was relatively low. Moreover, there was no specific effect on the face sheet. However, when the core pile height changed from sample group B (9.37 mm) to sample group C (7.46 mm) with the same face sheet thickness in both groups, the yield load was significantly decreased. Therefore, it can be concluded that after a specific core pile height, the flat compression performance of three-dimensional integrated woven spacer sandwich composite materials will continually decrease.

### 3.3. Tensile Performance

The weft tensile load–displacement curve and the maximum weft tensile load diagram are given in Figure 17 and Figure 18, respectively. As can be seen in Figure 17, the relationship between load and displacement of the four sample groups was linear. When the displacement reached a specific value, it entered a short buffer stage and maintained the linear relationship after buffering until broken. Based on this data, the ultimate tensile strength of three-dimensional integrated woven spacer sandwich composites can be calculated by the following formula:(4)$$F^{tu}=P^{\max}/A$$
where $F^{tu}$ is the ultimate tensile strength (MPa), $P^{\max}$ is the maximum load before failure (*N*), and *A* is the average cross-sectional area (mm^2^).

As can be seen in Table 7, the weft tensile strength gradually decreased with the increase in core pile height. However, when the face sheet thickness increased with the increase in core pile height in group D, the degradation became lower and almost similar to the tensile strength of group C samples. As can be seen in Figure 19 the XR-CT image showed that the bottom face sheet had more extensive damage than the top face sheet. The microfiber tears in the top face sheet were uniform. In contrast, the bottom face sheet fibers split and twisted when tensile load was applied. The different fracture phenomenon in the top and bottom face sheet was due to the variation in the face sheet thickness.

Figure 20 gives the warp tensile load–displacement curve. It can be seen that the relationship between the load and displacement of the three specimens was linear. There was no significant dissimilarity between the warp and weft tensile load–displacement curves. The average maximum load for warp direction samples was 3625 N, and the ultimate tensile strength in the warp direction of sample group A was 42.65 MPa. Therefore, it can be concluded that the core pile height has the greatest influence on the tensile strength of three-dimensional integrated woven spacer sandwich composites. The face sheet thickness and fiber strength also influence the failure behavior and tensile strength.

XR-CT images were used to visualize the microstructural details and fracture morphologies of the fractured tensile specimen. XR-CT is an excellent tool for the characterization of three-dimensional structure as it detects internal defects and visualizes every cross-sectional view without cutting the actual sample [27,28,29]. The internal and external 3D microstructure of the tested tensile specimen and the fractures in different planes (XY, YZ, XZ) were noticeable in the tomogram. Multiple fracture mechanisms could be observed, i.e., microfiber tears, fibers twist and split, matrix, and fiber interfacial debonding and fiber pull out. Apart from the specific fracture, other parts of the specimen had no apparent damage or structure deformation.

### 3.4. Edgewise Compression Performance

As can be seen in Figure 21, different samples had different load–displacement relationships. Some samples showed a linear relationship until the load reached the maximum, some samples produced a large displacement at the beginning with a small load, and some samples fluctuated up and down until the sample began to break. The average maximum weft compression load was calculated by processing the original data, as shown in Figure 22, and the following formula was used to calculate the ultimate compression strength:(5)$$F^{cu}=p^{\max}/A$$
where $F^{cu}$ is the compression strength (MPa), $P^{\max}$ is the maximum load before failure (*N*), and *A* is the cross-sectional area at the test section (mm^2^).

From Table 8, it can be seen that the core pile height had a significant effect on the weft compression performance of the three-dimensional integrated woven spacer sandwich composites. The weft compression strength gradually decreased when the core pile height increased.

The warp load–displacement curve of sample group C can be seen in Figure 23. The average maximum load in the warp load–displacement curve for the group C sample was 9100 N. The maximum compression strength was 69.81 MPa, which was six times higher than the weft compression value of sample group C. Other researchers have shown that core pile height significantly affects the warp compression performance and that face sheet rupture and dislocation dominate the failure of warp compression samples [30]. According to this result, it can be concluded that the warp compression load-bearing capacity of three-dimensional woven sandwich composites is higher than the weft direction [31]. Due to the structural difference in the warp and weft direction samples, the structure and arrangement of the warp and weft direction yarn can significantly increase or decrease the edgewise compression performance of three-dimensional integrated woven spacer sandwich composites.

Figure 24 gives the XR-CT image of warp compression fractured specimen. It can be seen that the damage in the bottom face sheet was more evident than the top face sheet. The warp yarn was torn, while the weft yarn was undamaged. Therefore, it is clear that the face sheet thickness and the warp and weft direction pile arrangement have a significant effect on the failure behavior of three-dimensional integrated woven spacer sandwich composites.

## 4. Conclusions

In this study, three-dimensional integrated woven spacer sandwich composites with five different specifications were developed by high-performance alkali-free E glass fiber and epoxy resin using the hand lay-up technique. XR-CT images were used to visualize the microstructure and fracture morphologies of the three-dimensional integrated woven spacer sandwich composites. The results showed that the core height, skin thickness, fiber strength, warp and weft structure, and resin impregnation significantly influenced the mechanical properties and performance of 3D integrated woven spacer sandwich composites. The bending result demonstrated that a low core height structure had better bending strength due to the “8” shaped structure and the action of the resin matrix. Meanwhile, the bending strength and antideformation ability were more significant for the weft direction than for the warp direction. The core height had a significant influence over the skin thickness on the flatwise compression and edgewise compression properties. The compression properties decreased with the increase in core height. The warp compression properties of the 3D integrated woven spacer sandwich composites were higher than the weft compression properties. Fiber strength and face sheet thickness had a considerable effect on tensile failure and properties. Moreover, the core pile height had a dominant effect on the tensile properties and performance of warp and weft direction structures. This research can help determine the mechanical properties and structure of three-dimensional integrated woven spacer sandwich composite materials for end-use products. A study comparing the fracture behavior of samples of different specifications under the same load can be conducted using XR-CT in order to determine the strength of composite materials.

## Figures and Tables

**Figure 1 materials-14-04284-f001:**
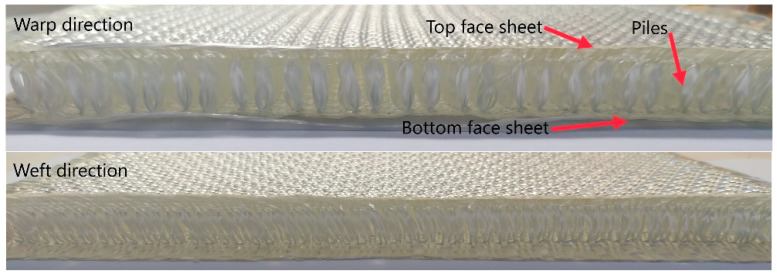
“8” shaped woven spacer fabric.

**Figure 2 materials-14-04284-f002:**
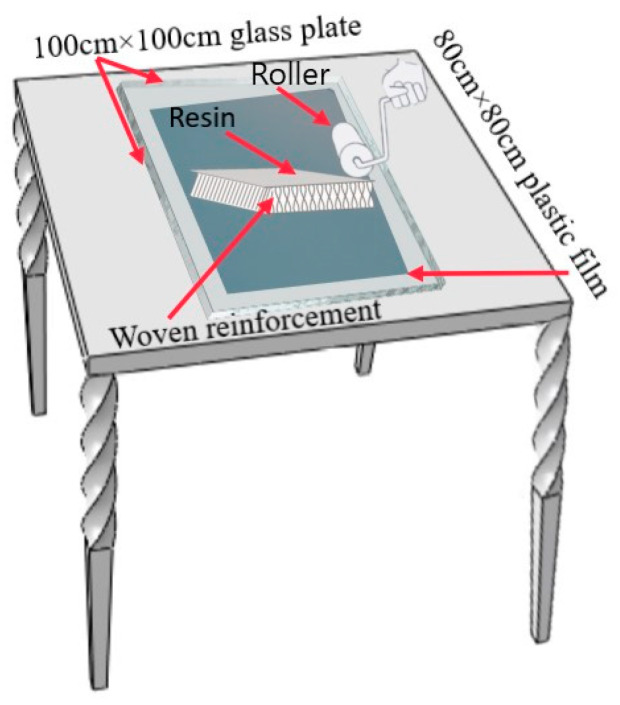
Schematic diagram of the hand lay-up process.

**Figure 3 materials-14-04284-f003:**
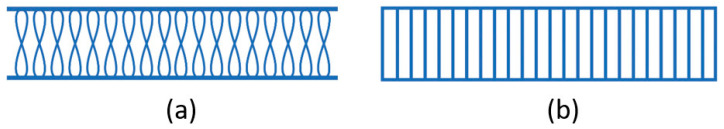
Schematic diagrams of (**a**) warp and (**b**)weft direction samples.

**Figure 4 materials-14-04284-f004:**
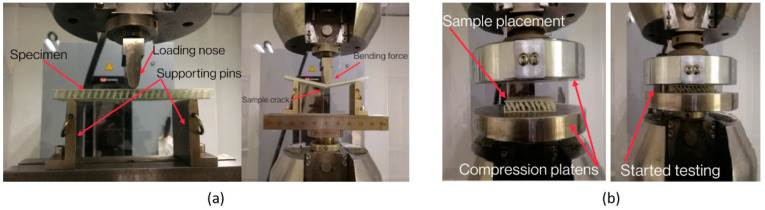
Equipment set ups for (**a**) bending and (**b**) flat compression experiments.

**Figure 5 materials-14-04284-f005:**
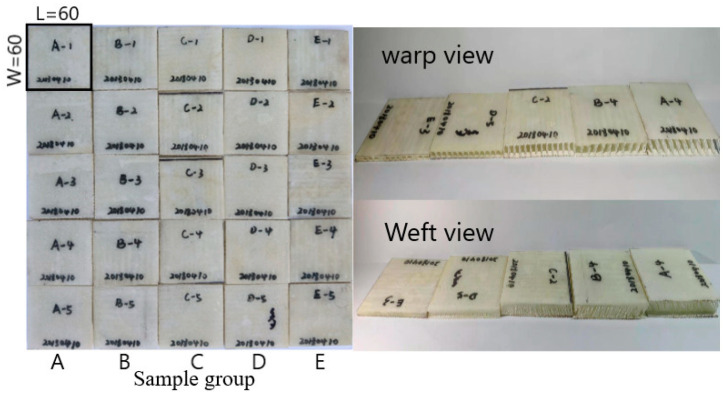
Picture of flat compression specimens.

**Figure 6 materials-14-04284-f006:**
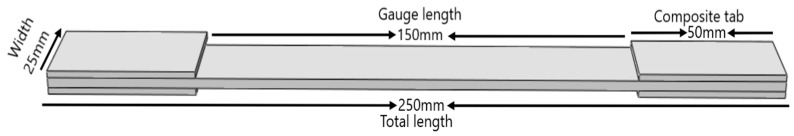
Schematic diagram of the tensile test specimen.

**Figure 7 materials-14-04284-f007:**
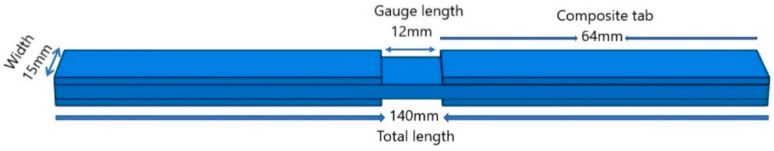
Schematic diagram of the edgewise compression test specimen.

**Figure 8 materials-14-04284-f008:**
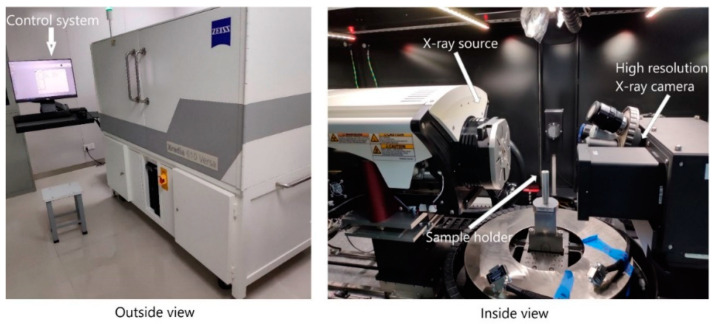
X-ray computed tomography machine.

**Figure 9 materials-14-04284-f009:**
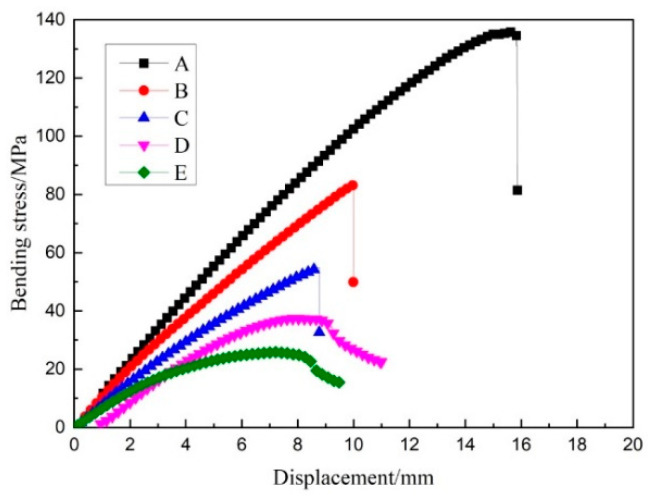
Warp stress–displacement curve.

**Figure 10 materials-14-04284-f010:**
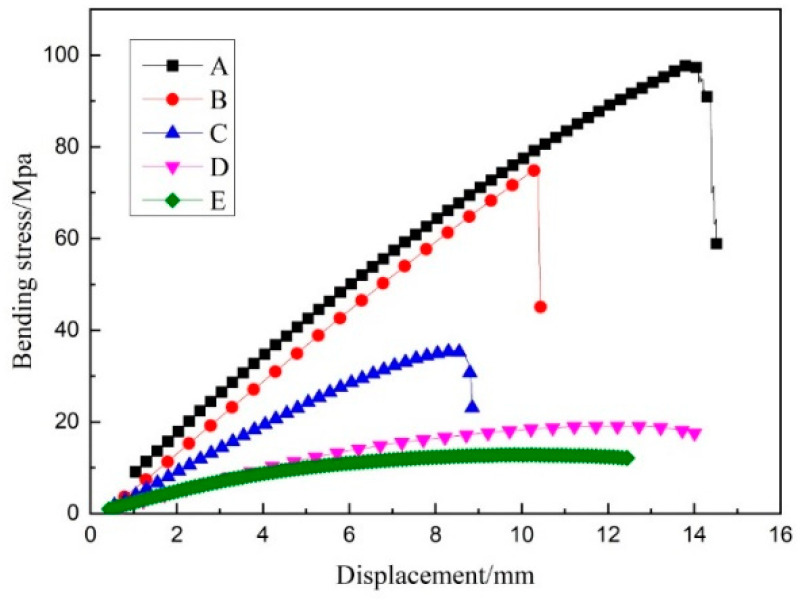
Weft stress–displacement curve.

**Figure 11 materials-14-04284-f011:**
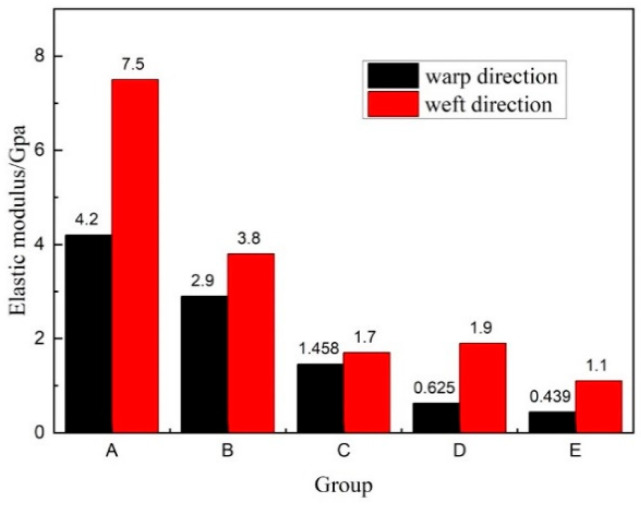
Average elastic modulus in warp and weft directions.

**Figure 12 materials-14-04284-f012:**
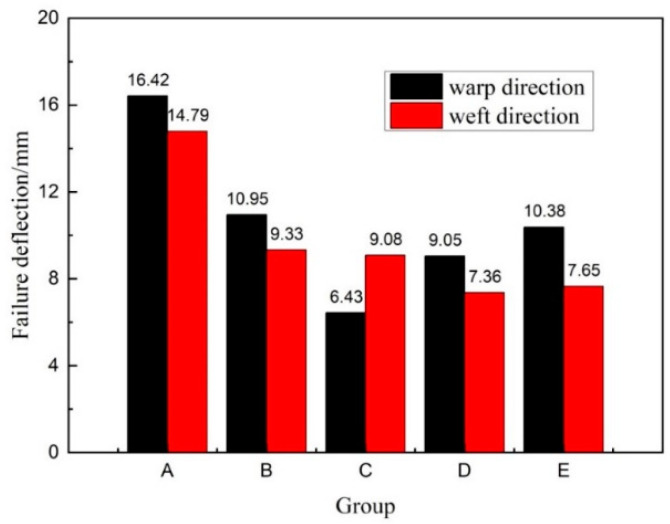
Failure deflection in warp and weft directions.

**Figure 13 materials-14-04284-f013:**
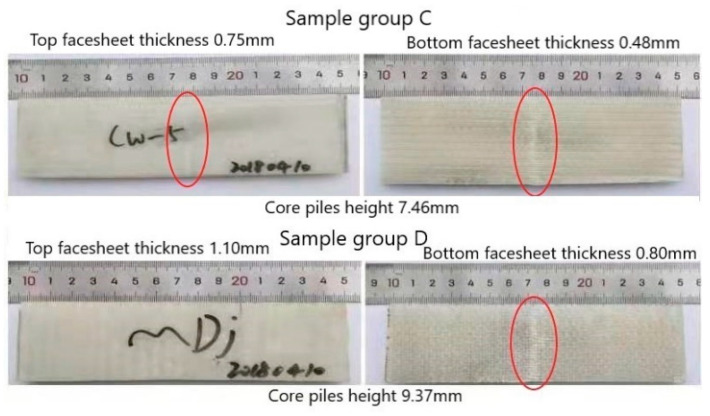
Reverse view of the damaged samples.

**Figure 14 materials-14-04284-f014:**
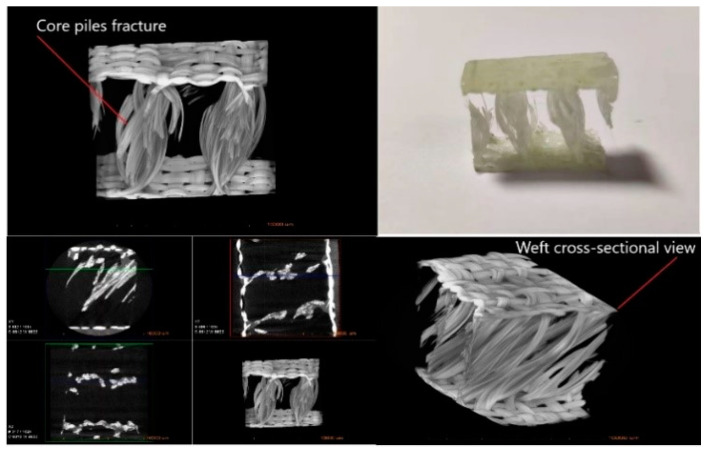
X-ray computed tomography images of the flexural fractured specimen.

**Figure 15 materials-14-04284-f015:**
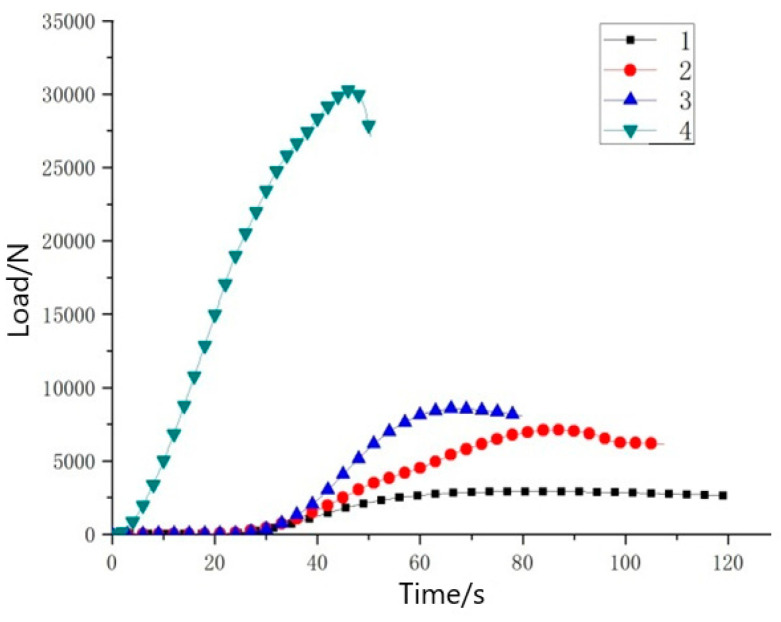
Sample load–time diagram.

**Figure 16 materials-14-04284-f016:**
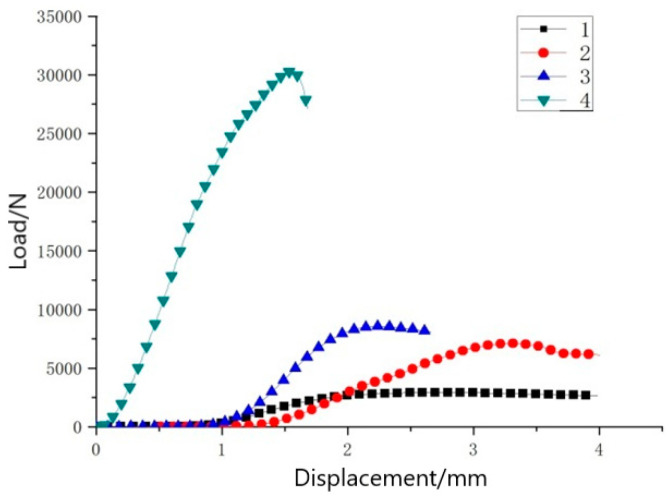
Sample load–displacement diagram.

**Figure 17 materials-14-04284-f017:**
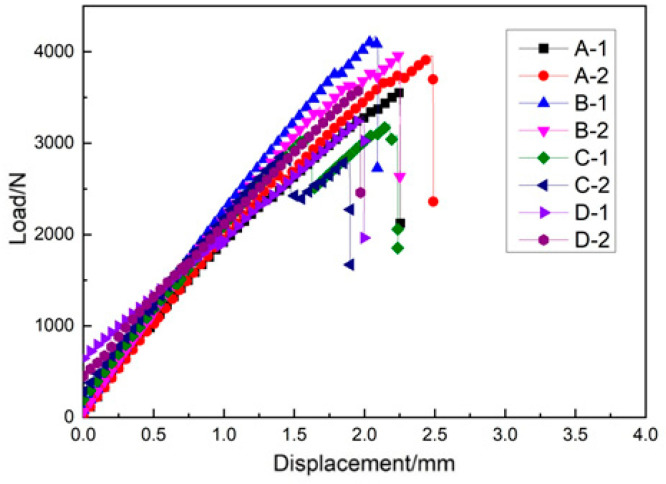
Weft tensile load–displacement curve.

**Figure 18 materials-14-04284-f018:**
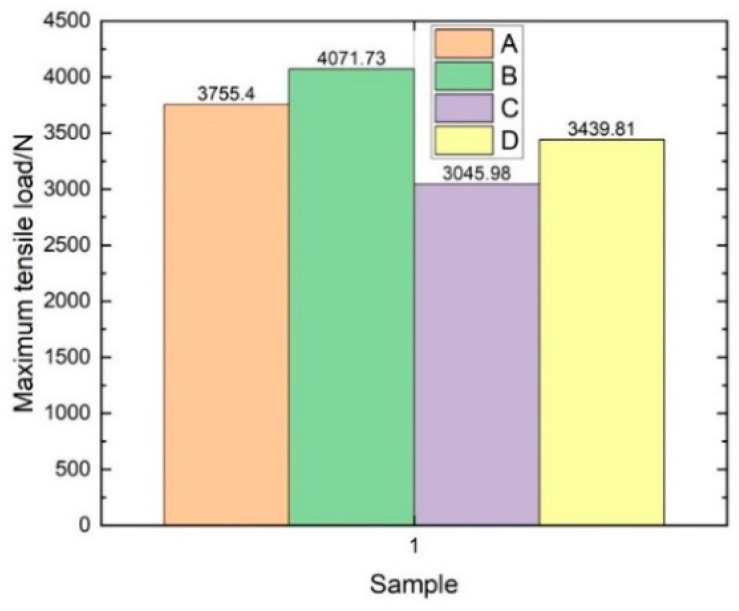
Maximum weft tensile load diagram.

**Figure 19 materials-14-04284-f019:**
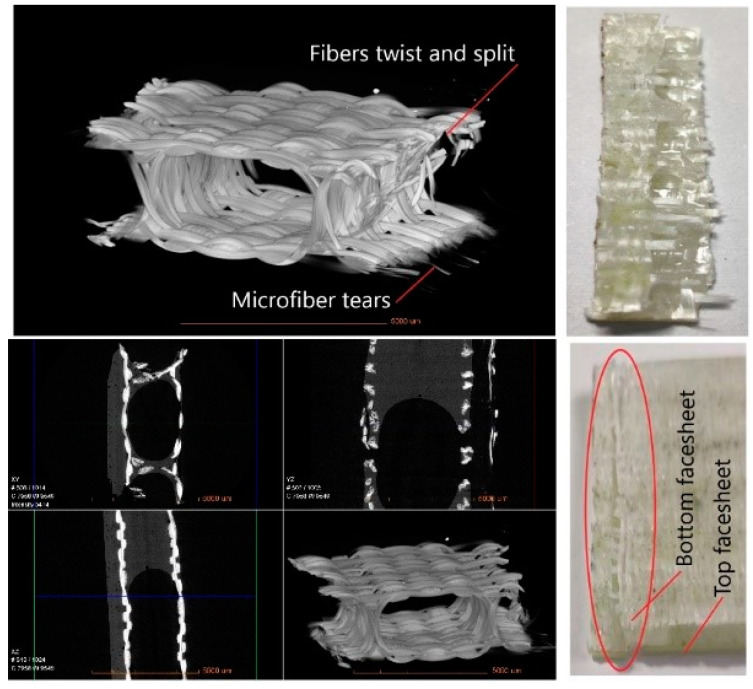
X-ray computed tomography image of the fractured tensile specimen.

**Figure 20 materials-14-04284-f020:**
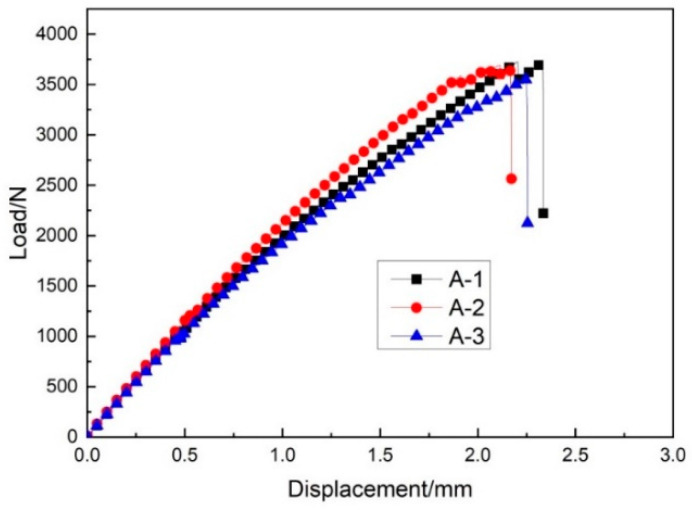
Warp tensile load–displacement curve of sample group A.

**Figure 21 materials-14-04284-f021:**
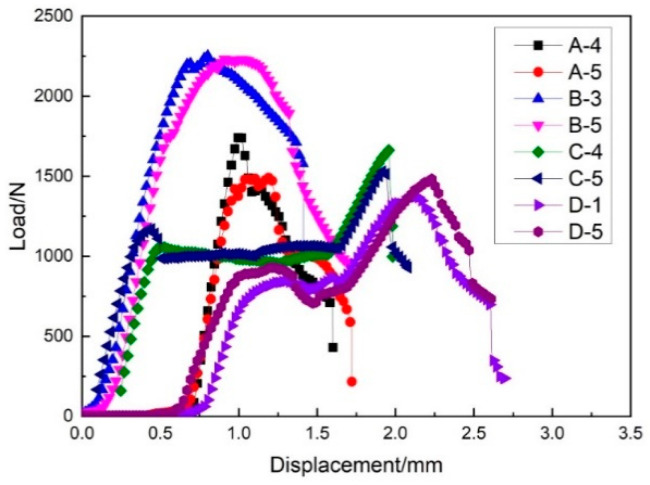
Load–displacement curve of weft compression.

**Figure 22 materials-14-04284-f022:**
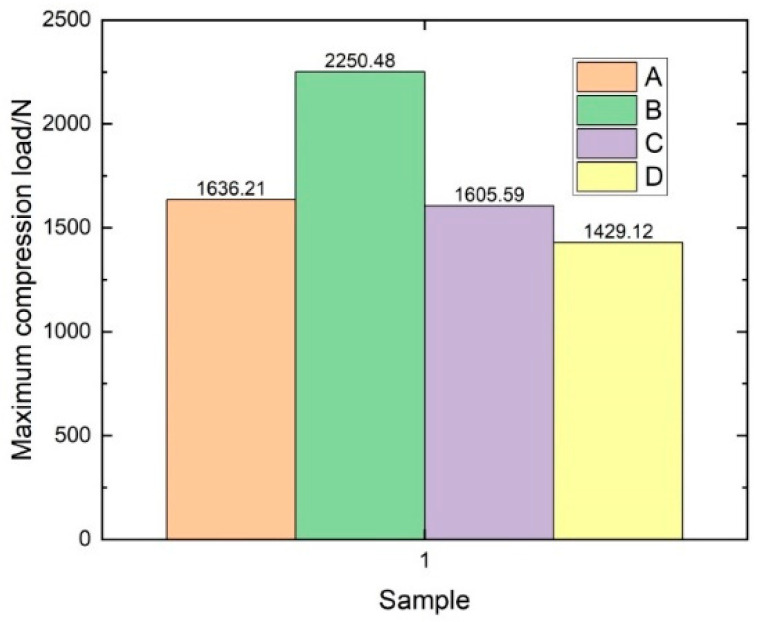
Maximum weft compression load diagram.

**Figure 23 materials-14-04284-f023:**
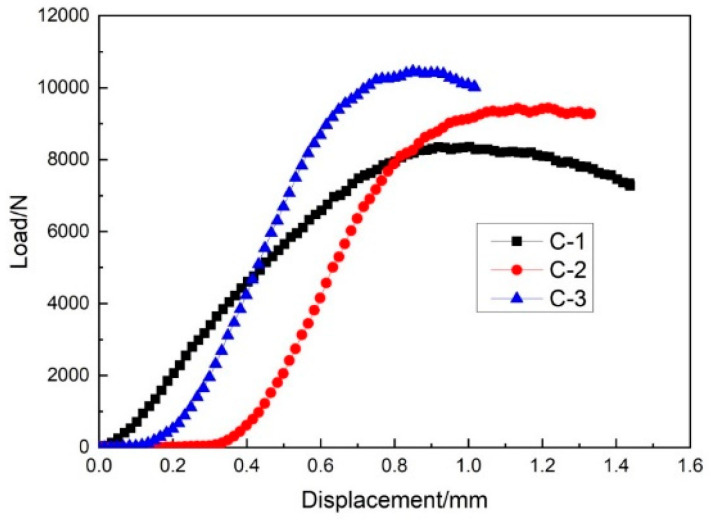
Warp compression load–displacement curve of sample C.

**Figure 24 materials-14-04284-f024:**
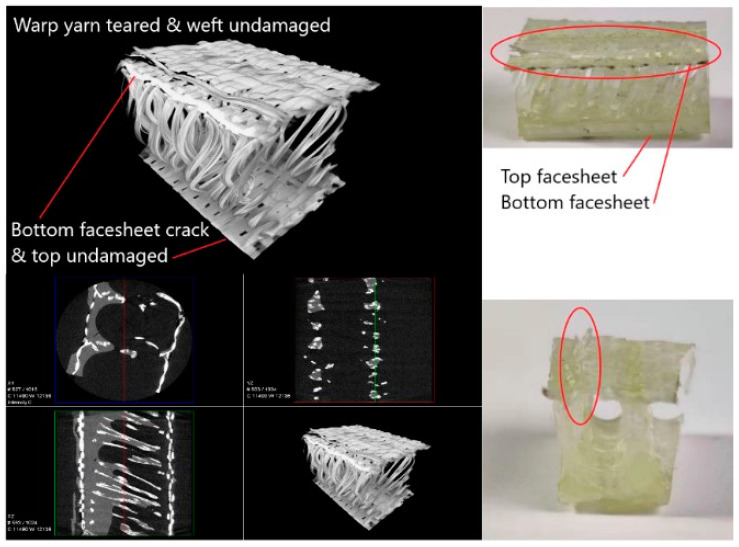
3D X-ray microscopy image of warp compression fractured specimen.

**Table 1 materials-14-04284-t001:** 3D integrated woven spacer fabric specifications.

Specifications	A	B	C	D	E
Core height (mm)	3.0	5.0	8.0	10.0	12.0
Face sheet thickness (mm)	0.36	0.36	0.36	0.58	0.58
Fabric weight (g/m^2^)	720	810	920	1440	1440

**Table 2 materials-14-04284-t002:** Specifications of 3D integrated woven spacer sandwich composite material.

Specifications (Average)	A	B	C	D	E
Total height (mm)	3.40	5.42	8.69	10.60	12.62
Core height (mm)	2.17	4.19	7.46	9.37	11.39
Top face sheet thickness (mm)	0.75	0.75	0.75	1.10	1.10
Bottom face sheet thickness (mm)	0.48	0.48	0.48	0.80	0.80
Composite weight (g/m^2^)	1510	1680	1900	2930	3060

**Table 3 materials-14-04284-t003:** Two sets of samples for comparison.

Group One	Top Face Sheet Thickness (mm)	Bottom Face Sheet Thickness (mm)	Core Material Height (mm)	Group Two	Top Face Sheet Thickness (mm)	Bottom Face Sheet Thickness (mm)	Core Material Height (mm)
Sample group A	1.10	0.80	11.39	Sample group C	0.75	0.48	7.46
Sample group B	1.10	0.80	9.37	Sample group D	0.75	0.48	4.19
-	-	-	-	Sample group E	0.75	0.48	2.17

**Table 4 materials-14-04284-t004:** Average bending strength in warp and weft directions.

Group	Average Bending Strength in Warp Direction (MPa)	Average Bending Strength in Weft Direction (MPa)
A	94	144
B	80	78
C	32	45
D	20	50
E	14	27

**Table 5 materials-14-04284-t005:** Data of sample group A.

	Length (mm)	Width (mm)	Yield Load (N)	Compressive Stress at Yield (MPa)
1	61.28	59.96	2839	0.72
2	60.98	60.58	2769	0.75
3	60.60	60.59	3320	0.91
4	60.56	60.30	2773	0.76
5	60.02	59.88	2968	0.83
Average value	60.68	60.26	2934	0.80
Standard deviation	0.48	0.33	230.44	0.07
Coefficient of Variation	0.78	0.55	7.85	8.28

**Table 6 materials-14-04284-t006:** Data of sample group D.

	Length (mm)	Width (mm)	Yield Load (N)	Compressive Stress at Yield (MPa)
1	60.12	60.00	29046	8.07
2	61.26	60.34	33472	9.06
3	60.82	60.40	27999	7.62
4	60.68	60.62	30266	8.23
5	60.78	60.54	28597	7.77
Average value	60.72	60.38	29876	8.15
Standard deviation	0.45	0.24	2175	0.56
Coefficient of Variation	0.75	0.40	7.2	6.87

**Table 7 materials-14-04284-t007:** Ultimate tensile strength of four groups of weft samples.

Sample Group	Maximum Load (*n*)	Cross-Sectional Area (mm^2^)	Ultimate Tensile Strength (MPa)
A	3755.4	85	44.18
B	4071.73	135.5	30.05
C	3045.98	217.25	14.02
D	3439.81	265	12.98

**Table 8 materials-14-04284-t008:** The weft compression strength of four groups of samples.

Sample Group	Maximum Load (*N*)	Cross-Sectional Area (mm^2^)	Compression Strength (MPa)
A	1636.21	51	32.08
B	2250.48	81.3	27.68
C	1605.59	130.35	12.32
D	1429.12	159	8.99

## Data Availability

Data sharing not applicable.

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
