# Peer review of "Effect of Structural Differences on the Mechanical Properties of 3D Integrated Woven Spacer Sandwich Composites"

_materials, 2021, doi:10.3390/ma14154284_

Round 1
Reviewer 1 Report
In this manuscript authors show the results of a research work related to the comparation of different mechanical properties of composite materials based on impregnated woven fabrics. In this sense two different spacers have been studied: warp and weft directioned.
It is an interesting study but a deep revision must be performed before considering acceptance:
- English writing must be reviewed and improved
- No value of mechanical properties is accompanied by the measurement error (i-e. standard deviation). Standard deviation or other error measurement must be included. Moreover, no statistical study of the differences in values has been carried out in order to give support to the significant differences. Suggestion: performance of A-NOVA calculation of the variance.
- Authors inform about the chemical nature of the resin (epoxy) but it is first mentioned in conclusion’s section. It must be included in the materials section with a deeper description of the chemical structure of the polymeric compound.
- Information of Figure 3 and figure 4 could be included in just one figure, reflecting A and B.
- XRM microscopy should be referred as XR-CT (X-ray computed tomography)
- A deep discussion of the XR-CT images must be performed in order to complete thedescription of the mechanical results.
Author Response
Comments and Suggestions for Authors
In this manuscript authors show the results of a research work related to the comparation of different mechanical properties of composite materials based on impregnated woven fabrics. In this sense two different spacers have been studied: warp and weft directioned.
It is an interesting study but a deep revision must be performed before considering acceptance:
- English writing must be reviewed and improved
Authors response: English writing have reviewed and improved throughout the manuscript as required.
- No value of mechanical properties is accompanied by the measurement error (i-e. standard deviation). Standard deviation or other error measurement must be included. Moreover, no statistical study of the differences in values has been carried out in order to give support to the significant differences. Suggestion: performance of A-NOVA calculation of the variance.
Authors response: Average value, standard deviation and coefficient of variations was just calculated to figure out the amount variations of each data from the average value, it was not used to support mechanical properties or performances evaluation, just used for knowing the measurement difference from one samples to others. Therefore, calculate statistical model may not necessary for it.
- Authors inform about the chemical nature of the resin (epoxy) but it is first mentioned in conclusion’s section. It must be included in the materials section with a deeper description of the chemical structure of the polymeric compound.
Authors response: Specifications of resin matrix have added in the materials section, can be seen in line numbers 80,81,82 and 83.
- Information of Figure 3 and figure 4 could be included in just one figure, reflecting A and B.
Authors response: Figure 3 and 4 have merged together and represented by (a) and (b). Moreover, next all figures number arranged serial wise.
- XRM microscopy should be referred as XR-CT (X-ray computed tomography)
Authors response: XRM microscopy has referred as XR-CT in the whole manuscript.
- A deep discussion of the XR-CT images must be performed in order to complete the description of the mechanical results.
Authors response: Required discussion about XR-CT images have added in line numbers 253-258 and 346-351.

Reviewer 2 Report
Dear Authors,
your article is interesting and focused on the journal scope. However, I suggest to spend additional efforts to improve its overall quality:
- abstract should include additional considerations: it is quite vague.
- the introduction should be also improved: it is also rather vague, not able to define the state of the art
- in the introduction, please, include additional papers and explain them
- the introduction misses to explain the originality/value of your paper
- methods & materials are ok, but pictures (there and elsewhere) have to be improved. For instance labels can be hardly understood.
- Results have to be better explained and discussed.
- Comparisons with results from other authors are welcome.
- the general discussion has to be reinforced.
- an intense correction of language has to be implemented.
- journal format has to be doublechecked too
- some minor aspects are provided in the attached file

Author Response
Comments and Suggestions for Authors
Dear Authors,
your article is interesting and focused on the journal scope. However, I suggest to spend additional efforts to improve its overall quality:
- abstract should include additional considerations: it is quite vague.
Authors response: Abstract have modified and additional considerations have included.
- the introduction should be also improved: it is also rather vague, not able to define the state of the art
Authors response: Introduction part have improved and separated in three different part for clear understanding.
- in the introduction, please, include additional papers and explain them
Authors response: Two others references have added and described in the introduction section, can be seen in line numbers 52-58.
- the introduction misses to explain the originality/value of your paper
Authors response: Innovations of this research briefly discussed in the last part of the introduction.
- methods & materials are ok, but pictures (there and elsewhere) have to be improved. For instance labels can be hardly understood.
Authors response: Figure 3 and figure 4 have modified.
- Results have to be better explained and discussed.
Authors response: Results and discussion part have improved, modifications can be seen through track changes.
- Comparisons with results from other authors are welcome.
- the general discussion has to be reinforced.
Authors response: Discussion part have improved, modifications can be seen through track changes.
- an intense correction of language has to be implemented.
Authors response: Writings have reviewed and improved throughout the manuscript.
- journal format has to be doublechecked too
Authors response: Template have downloaded from the journal website and for revision manuscript have downloaded from the recommended link.
- some minor aspects are provided in the attached file
Authors response: Correction have done for all the minor aspects as suggested.

Reviewer 3 Report
The paper focuses on the experimental characterisation of the mechanical behaviour (bending, compressive and tensile solicitation) of 3D woven spacer sandwich composites.
In my opinion the title does not reflect totally the aim of the paper and should be improved/changed.
Overall, the subject of the paper is original and in line with the scope and aim of the journal. The state of the art is reviewed in acceptable way. The Introduction needs to be enriched. The presentation and discussion of results is often not easy to understand. Particularly, the English needs being improved and enhanced, some text has to be rephrased to be more readable (see comments below).
In my opinion the paper can be accepted after the following points are amended (major revision):
- Introduction: The originality of the contribution is not sufficiently discussed. The aim of the paper is not clearly presented.
- The English should be enhanced: some phrases are awkward and should be modified. For instance, already the title “Effect of structural difference on the mechanical properties of 3D woven spacer sandwich composite” is quite odd and difficult to understand. In Section 3 Results and Discussion some phrases are truly odd and need being rephrased. I suggest re-reading from an English native speaker.
Author Response
Comments and Suggestions for Authors
The paper focuses on the experimental characterisation of the mechanical behaviour (bending, compressive and tensile solicitation) of 3D woven spacer sandwich composites.
In my opinion the title does not reflect totally the aim of the paper and should be improved/changed.
Authors response: Some changes have done in the title. In the whole manuscript discussed about the mechanical properties and structural changes effect of 3D integrated woven spacer sandwich composites, so the aim of the paper is clear now.
Overall, the subject of the paper is original and in line with the scope and aim of the journal. The state of the art is reviewed in acceptable way. The Introduction needs to be enriched. The presentation and discussion of results is often not easy to understand. Particularly, the English needs being improved and enhanced, some text has to be rephrased to be more readable (see comments below).
In my opinion the paper can be accepted after the following points are amended (major revision):
- Introduction: The originality of the contribution is not sufficiently discussed. The aim of the paper is not clearly presented.
Authors response: Introduction part have improved and the innovations of this research briefly discussed in the last part of the introduction.
- The English should be enhanced: some phrases are awkward and should be modified. For instance, already the title “Effect of structural difference on the mechanical properties of 3D woven spacer sandwich composite” is quite odd and difficult to understand. In Section 3 Results and Discussion some phrases are truly odd and need being rephrased. I suggest re-reading from an English native speaker.
Authors response: English writing have reviewed and improved throughout the manuscript as required.

Round 2
Reviewer 1 Report
The document can be accepted in the present form
Reviewer 2 Report
No further remarks
Reviewer 3 Report
The authors have satisfactorily replied to all my queries.
The paper can now be accepted for publication.